# Cost of implementing a doxycycline test-and-treat strategy for onchocerciasis elimination among settled and semi-nomadic groups in Cameroon

Guillaume Trotignon[1]*, Ruth Dixon[1], Kareen Atekem[2], Laura Senyonjo[1], Joseph Kamgno[3,4], Didier Biholong[5], Iain Jones[1], Rogers Nditanchou[2]

1 Sightsavers, Haywards Heath, United Kingdom, 2 Sightsavers, Cameroon Country Office, Yaoundé, Cameroon, 3 Centre de recherche sur la filariose et d'autres maladies tropicales (CRFilMT), Fouda Quarter, Yaoundé, Cameroon, 4 Faculty of Medicine and Biomedical Sciences, University of Yaoundé; Yaounde, Cameroon, 5 National Programme for the Control of Lymphatic Filariasis and onchocerciasis, Ministry of Public Health, Yaoundé, Cameroon

* gtrotignon@sightsavers.org

**Data Availability Statement:** All relevant data are within the paper and its Supporting Information files.

## Abstract

### Background

Onchocerciasis is a neglected tropical disease with 217.5 million people globally at risk of having the infection. In both settled and semi-nomadic communities of Massangam Health District in Cameroon, Sightsavers has been carrying out test-and-treat with doxycycline and twice-yearly ivermectin distribution. This paper focuses on the cost of test-and-treat with doxycycline in the two community contexts of settled and semi-nomadic.

### Methods

For the valuation, a combination of gross or micro-costing was used to identify cost components, as well as bottom-up and top-down approaches. The opportunity costs of vehicle and equipment use were estimated and included. Not included, however, were the opportunity costs of building use and Ministry of Public Health staff salaries. We only captured the incremental costs of implementing test-and-treat activities as part of a functional annual community-directed treatment with the ivermectin programme.

### Results

We estimate the economic cost per person tested and cost per person treated in Massangam to be US$135 and US$667 respectively. Total implementation cost in the settled community was US$79,409, and in the semi-nomadic community US$69,957. Overall, the total economic cost of implementing the doxycycline test-and-treat strategy for onchocerciasis elimination in Massangam came to US$168,345. Financial costs represented 91% of total costs.

**Funding:** The author(s) received no specific funding for this work.

**Competing interests:** The authors have declared that no competing interests exist.

## Conclusions

Unit costs of test-and-treat in both settled and semi-nomadic communities are higher than unit costs of community-directed treatment with ivermectin. However, it is critical to note that a two-year implementation shows a significantly larger reduction in infection prevalence than the preceding 20 years of annual community-directed treatment with ivermectin. Test-and-treat with doxycycline may be a cost-effective intervention in places where the prevalence of microfilaria is still high, or in hard-to-reach areas where community-directed treatment with ivermectin and MDA coverage are not high enough to stop transmission or where marginalised populations consistently miss treatment.

### Author summary

Onchocerciasis, also known as river blindness, is a neglected tropical disease (NTD) caused by a parasitic nematode, *Onchocerca volvulus*. It is transmitted to humans through the bites of infected blackflies of the genus *Simulium*, which breed in fast-flowing rivers and streams. To eliminate transmission, a yearly distribution of ivermectin is conducted in infected areas. This medication kills the microfilaria but not the adult nematode, which is killed by doxycycline. Despite over 20 years of ivermectin mass distribution, the disease is still prevalent in the Massangam Health District in Cameroon. Therefore, alternative strategies have been discussed to stop transmission, including treatment of positive tested persons with doxycycline in both settled and nomadic communities. This paper estimated the total cost of implementation at US$168,345 in Massangam district, in order to compare it with the current elimination strategy and to demonstrate the potential effectiveness of the intervention–especially in high infection, hard-to-reach and *loa loa* co-infection settings.

## Introduction

Onchocerciasis is a neglected tropical disease (NTD) caused by a parasitic nematode, *Onchocerca volvulus*. It is prevalent in rural and poor settings in Africa and is transmitted to humans through the bites of infected blackflies of the genus *Simulium*, which breed in fast-flowing rivers and streams.

Also known as river blindness, the disease results in stigma and disability due to skin lesions, severe itching and ocular disorders including visual impairment, which can lead to irreversible blindness [1,2]. In 2019, the World Health Organization (WHO) estimated that 217.5 million people globally are at risk of having the infection and that more than 99% of all onchocerciasis cases are in sub-Saharan Africa [3].

In the new WHO roadmap on NTDs 2021–2030, onchocerciasis has been targeted for elimination. However, there are geographical locations under WHO-recommended ivermectin mass drug administration (MDA) that continue to have persistently high transmission or unsatisfactory progress towards elimination [4]. The WHO has therefore recommended a series of different strategies in these areas to address these challenges and move towards elimination [5,6].

Massangam Health District in the West Region of Cameroon is one location not on track to eliminate onchocerciasis, despite receiving 20 years of uninterrupted annual distribution of ivermectin [7,8]. To redress this situation, Sightsavers (in collaboration with the national control programme), implemented test-and-treat with doxycycline (TTd) as part of a suite of alternative treatment strategies (ATS) that also included larviciding and biannual MDA [9].

In this paper, we estimate the cost of implementing a TTd programme in two community contexts: settled and semi-nomadic. We have focused on the second round of TTd that was implemented in September 2018 among the settled communities of Massangam, and between October 2019 and December 2019 among the semi-nomadic population living in encampments. We estimate the implementation cost and examine the cost drivers for each scenario.

Secondly, we have costed the two supplementary activities conducted in the semi-nomadic implementation: satellite nomadic camp verification and microscopic negative sample examined by polymerase chain reaction (PCR). These are optional activities and are referred to as 'add-ons' throughout this paper. We estimate the gains in terms of coverage of testing and treating positive cases through each add-on.

Finally, we discuss the implications of these costs in decision-making for treatment programme selection and implementation. We reflect on the trade-off between coverage and cost for the two add-ons, and we compare the findings with community-directed treatment with ivermectin (CDTi), including the impacts on disease prevalence, intensity and timeline to elimination. This analysis provides a foundation upon which future endeavours to eliminate onchocerciasis can build.

## Methods

### Setting

As mentioned above, Massangam Health District in the West Region of Cameroon is not on track to eliminate onchocerciasis. Alternative strategies were therefore conducted to address it, namely test-and-treat with doxycycline (TTd), larviciding and biannual MDA [7–9].

For the TTd component of the ATS, all eligible individuals (those aged eight years and older, who are not breastfeeding, not pregnant and not severely ill) were screened using skin-snip microscopy for onchocercal microfilarial (MF) infection. Those positive (presence of MF) were offered 100mg of doxycycline with a light meal daily for 35 days under directly observed therapy (daily observation) strategies by trained community drug distributors (CDDs) [9,10].

Two rounds of TTd were conducted among settled communities, the first in July 2017 and the second in September 2018, following a sequence of sensitisation, census, screening and treatment. Another round of TTd targeted semi-nomadic communities between October and December 2019 (Table 1).

During the first two rounds of ATS implementation, the presence and challenges of reaching semi-nomadic populations with TTd and ivermectin came to light. In response, adaptations to the strategies were developed and implemented, including culturally adapted mobilisation and sensitisation materials such as banners, flyers, posters, video and audiotapes in four local languages (Pidgin English, French, Bamoun and Fulfulde) used at national, regional, district and community levels.

In addition, mobilisation and sensitisation were extended and reinforced in every camp by two CDDs in a "buddied up" system–one from the settled community and another from the semi-nomadic community. For semi-nomads, daily monitoring was unfeasible, and treatment was monitored daily only for the first two days and then weekly, with two additional checks by phone each week.

Within the semi-nomadic communities, two further activities were piloted around the TTd with the aim of further increasing coverage. Firstly, there was the identification of semi-nomadic camps by satellite imagery. Verification on the ground of these camps' occupation (also called camp verification) was then undertaken to supplement census activities conducted via community networks. Secondly, skin snips found to be negative by microscopy were

**Table 1. Test-and-treat standard activities with adaptation and add-on for semi-nomadic groups.**

| Activities | First round settled communities (July 2017) | Second round settled communities (September 2018) | Semi-nomadic communities (October to December 2019) |
|---|---|---|---|
| **Standard activities** | | | |
| **Planning and coordination** | All activities related to operational, management and coordination of TTd activities. | | Same as previous column |
| **Advocacy and sensitisation** | Meeting with national and local stakeholders, sharing information about the TTd programme at local level. | | Culturally adapted mobilisation and sensitisation materials (video and audio in different languages). |
| **Training** | Training of field staff, including settled community CDDs. | | Training of semi-nomadic CDDs. |
| **Census** | Team goes into the community and goes door-to-door registering people and determining eligibility and sensitising for the upcoming test. | | Team goes to the actual camps after undergoing a community listing process, and snowballs at each camp to identify unlisted camps. They then go hut-to-hut registering people. |
| **Testing** | Skin snip and field examination under microscope. | | Same approach but a nomad community guide accompanies the team. |
| **Treatment and monitoring** | Another team follows up positive results, conducting pregnancy tests where appropriate before treating under DOTd. A meal is provided before medicine is ingested. | | DOTd[a] for two days, then a weekly treatment course, self-monitored in a treatment diary, and funds provided for the meals. Treatment team calls (twice) during the week and visits weekly to monitor treatment. Drug packages and diaries were reviewed to ascertain compliance during visits. |
| **Add-on activities** | | | |
| **Satellite nomadic camp verification** | Not implemented in settled communities. | | For nomadic camps, in addition to the community listing of camps and snowballing, potential camps were identified by satellite and visited to verify if they were occupied. |
| **PCR testing of negative skin snips from nomadic participants[b]** | Not implemented in settled communities. | | Due to the low sensitivity of microscopy, negative samples from semi-nomads were retained and further analysed by PCR. |

Legend: PCR (Polymerase Chain Reaction).

[a]DOTd (Directly Observed Treatment with doxycycline)

[b]*Add-ons will be presented separately as they are not standard TTd activities*

examined by PCR in the Centre for Research in Filaria and other Tropical Diseases (CRFilMT) laboratory in Yaoundé. This was to identify further positive cases for treatment [8,9].

## Costing approach

The interventions were jointly implemented by the Ministry of Public Health of Cameroon, Sightsavers and CRFilMT in three communities (Njinjia/Njngouet, Makakoun and Makouop-sap) in the Massangam Health District where the infection rate was much higher than the elimination threshold of 0.1% [2,7]. Different measurement and cost valuation methodologies were used as the level of necessary detail was not obtained to accurately estimate the cost of several activities (see S1 Table). This study estimates the cost under the perspective of the service providers (Ministry of Public Health of Cameroon, CRFilMT and Sightsavers) to conduct one phase of TTd among settled groups and one phase among semi-nomadic groups.

Following existing guidelines, a mix of gross or micro-costing was used to identify cost components, as well as bottom-up and top-down approaches, for their valuation [11]. Standard costs are costs of activities that, based on project experience, are deemed necessary for an optimal implementation. The opportunity costs of vehicle and equipment use were estimated and included, while the value of volunteered time of CDDs was evaluated using the opportunity cost approach [12]. As pilot activities took place in a rural area, we used the yearly "agriculture, forestry and fishing, value added per worker" from the World Bank dataset as a shadow wage to value the daily work of CDDs [13]. However, opportunity costs of building

use and salaries of staff not specifically employed for the project were not included, as information on time spent and salary levels could not be retrieved. We only captured the incremental costs of implementing TTd activities as part of a functional annual CDTi programme.

### Data collection

In 2020, actual and budgeted expenditure for conducting ATS activities were collected retrospectively from implementing partners and the Ministry of Public Health. Sightsavers' expenditures were downloaded as a transaction list from the financial tracking system and consolidated via Excel spreadsheets.

### Data analysis

The delivery of TTd includes several activities, some of which were adapted or added to suit the nomadic community context. Details of the activities are given in Table 1 above.

Project costs were collected or valued, then categorised into specific project activities and cost categories to identify cost drivers and unit costs (see S2 Table). All capital expenditure and equipment with usefulness exceeding one year were annualised assuming an average life of five years and a discount rate of 3%, following WHO guidelines [14,15].

The total standard cost of implementing a TTd strategy was estimated based on the project output and financial data. The cost by activity and category are presented, with and without add-ons, along with estimates of the cost per person screened and cost per person treated, for both settled and semi-nomadic communities (see S1 Dataset).

### Currency

Following the Turner et al. first adjustment method, all costs were converted to US dollars (USD) from CFA francs (XAF) or UK pounds (GBP) using the average monthly exchange rate for the year 2020, and adjusted for inflation to USD 2020 using consumer price indexes [16–18]. This method was used as personnel unit costs did not evolve between 2019 and 2020, and most equipment and supplies were imported tradable goods.

### Results

The total cost of test-and-treat activities in settled communities was US$79,409 and the total cost less overheads was US$69,051 (see Table 2). Planning and coordination activities

**Table 2. Cost of test-and-treat activities in settled communities by activities and categories (in USD 2020 and percentage of total cost without overheads).**

| Activities/ categories | Personnel | Materials and supplies | Transportation | Total |
|---|---|---|---|---|
| **Planning and coordination** | 21,205.1 (31%) | 27.0 (0%) | 2,343.6 (3%) | 23,575.6 (34%) |
| **Advocacy and sensitisation** | 2,943.4 (4%) | 2,681.1 (4%) | 1,771.7 (3%) | 7,396.2 (11%) |
| **Training** | 617.3 (1%) | - | 60.0 (0%) | 677.3 (1%) |
| **Census** | 1,063.6 (2%) | 70.1 (0%) | 1,597.9 (2%) | 2,731.5 (4%) |
| **Testing** | 1,401.6 (2%) | 6,327.0 (9%) | 2,752.8 (4%) | 10,481.4 (15%) |
| **Treatment** | 4,213.0 (6%) | 12,634.7 (18%) | 3,279.8 (5%) | 20,127.5 (29%) |
| **Monitoring of test-and-treat** | 1,797.5 (3%) | - | 2,264.2 (3%) | 4,061.7 (6%) |
| **Sub-total** | 33,241.5 (48%) | 21,739.8 (31%) | 14,069.9 (20%) | 69,051.3 (100%) |
| Overheads[a] | - | - | - | 10,357.7 (13%) |
| Total | - | - | - | 79,409.0 (100%) |

[a]Overheads estimated at 15% of sub-total activity costs

**Table 3. Cost of test-and-treat activities in semi-nomadic communities by activities and categories (without add-ons, in USD 2020 and percentage of total cost without overheads).**

| Activities/ categories | Personnel | Materials and supplies | Transportation | Total |
|---|---|---|---|---|
| **Planning and coordination** | 9,154.4 (15%) | 2,152.2 (4%) | 175.2 (0%) | 11,481.8 (19%) |
| **Advocacy and sensitisation** | 5,431.3 (9%) | 3,781.8 (6%) | 2,084.9 (3%) | 11,298.0 (19%) |
| **Training** | 1,024.3 (2%) | 447.7 (1%) | 262.8 (0%) | 1,734.8 (3%) |
| **Census** | 3,310.7 (5%) | 1,770.6 (3%) | 3,027.5 (5%) | 8,108.9 (13%) |
| **Testing** | 2,942.9 (5%) | 7,885.9 (13%) | 3,588.2 (6%) | 14,277.3 (23%) |
| **Treatment** | 1,441.5 (2%) | 1,238.7 (2%) | 1,252.5 (2%) | 3,932.7 (6%) |
| **Monitoring of test-and treat** | 4,007.9 (7%) | - | 5,851.2 (10%) | 9,859.1 (16%) |
| **Sub-total** | 27,313.1 (45%) | 17,276.7 (28%) | 16,242.3 (27%) | 60,832.1 (100%) |
| **Overheads**[a] | - | - | - | 9,124.8 (13%) |
| **Total** | - | - | - | 69,956.9 (100%) |

[a]Overheads estimated at 15% of sub-total activity costs

represented the highest share of costs (34% of total cost less overheads, US$23,576), followed by treatment activities (29%, US$20,127). Overall, personnel represented the highest cost category (48% of total expenditure) followed by materials and supplies (31%).

In semi-nomadic communities, the total cost of the adapted test-and-treat activities was US $69,957 and US$60,832 without overheads (Table 3). In terms of activity, testing was the highest proportion of costs (23% of total, US$14,277). A significant share of this (US$7,886) was due to materials and supplies being needed on site by the testing team in order to camp and conduct sample collection and examination. Personnel represented the highest cost category with 45% of the total cost.

The overall economic cost of implementing a TTd with a tailored strategy to reach settled and semi-nomadic groups in Massangam amounted to US$149,366 with a cost per person screened of US$135 and a cost per person treated of US$667 (Table 4).

A cost difference is to be noted when looking separately at the implementation among settled and semi-nomadic communities. TTd strategy cost US$79,409 among the settled communities, while cost among semi-nomadic communities was US$69,957. In terms of unit cost per person tested and treated, this corresponds to US$90 and US$389 respectively in settled communities, and US$310 and US$3,498 for semi-nomadic groups without add-ons; and US$394 and US$4,447 with add-on activities. The financial costs for both community settings represented most of the total expenses (91%). The additional economic costs correspond mostly to the estimated wage of CDDs among settled communities and to vehicle hire among semi-nomadic communities.

**Table 4. Total economic cost per community and per outcome (in USD 2020)[a].**

| | Number of persons censused | Number of persons screened | Number of persons treated | Total economic cost[b] | Cost per person censused | Cost per person screened | Cost per person treated |
|---|---|---|---|---|---|---|---|
| Settled communities | 1,617 | 881 | 204 | 79,409 (73,262) | 49 (45) | 90 (83) | 389 (359) |
| Semi-nomadic communities | 748 | 226 | 20 | 69,957 (62,978) | 94 (84) | 310 (29) | 3,498 (3,149) |
| Total Massangam (without add-ons) | 2,365 | 1,107 | 224 | 149,366 (136,240) | 63 (58) | 135 (13) | 667 (608) |

[a]Financial cost in brackets

[b]Including overheads

**Table 5. Incremental cost and outcomes of add-ons among semi-nomadic communities (USD 2020)[a].**

| | Additional persons censused | Additional persons screened | Additional persons treated | Total economic cost (Financial cost)[b] | Cost per person censused | Cost per person screened | Cost per person treated |
|---|---|---|---|---|---|---|---|
| Camp verification | 11 | 11 | - | 8,709 (8,543) | 792 (777) | 792 (777) | - |
| PCR analysis | - | 161 | - | 10,270 (10,270) | - | 64 (64) | - |
| Additional treatment[c] | - | - | 14 | 3,166 (2,744) | - | - | 197 (170) |
| Total add-ons | 11 | 172 | 14 | 22,145 (21,557) | 2,013 (1,960) | 129 (125) | 1,581 (1,540) |

Legend: PCR (Polymerase Chain Reaction)

[a]Financial cost in brackets

[b]Including overheads

[c]Additional treatment costs extrapolated from standard activities but not included in total costs as treatment was not conducted immediately due to COVID related restrictions. Presented here for readers' information and to present the number of patients infected detected due to add-on activities (Table 4)

The economic cost of add-on activities amounted to US$18,979 (excluding the cost of additional treatment, already included above), increasing the total TTd programme cost by 13% (Table 5). This corresponds to US$129 per additional person screened and US$1,581 per person treated. Financial costs represented 98% of total economic costs. The camp verification helped in detecting two more camps and 11 additional persons. PCR tests resulted in detecting 14 infections from 161 tests conducted among semi-nomadic camps [10].

## Discussion

This study shows how much alternative treatment strategies would cost in both settled communities and semi-nomadic communities in hard-to-reach areas. To the best of our knowledge, this is the first cost estimate of onchocerciasis treatment among semi-nomadic groups. Based on financial and output data, we estimate an economic cost per person tested and cost per person treated in Massangam to be US$135 and US$667 respectively. Total implementation cost in the settled community was US$79,409 and in the semi-nomadic community US$69,957. Overall, the total cost of implementing the doxycycline test-and-treat strategy for onchocerciasis elimination in Massangam came to US$168,345.

Unit costs are estimated to be higher among semi-nomadic groups compared to unit costs among settled groups. This is because implementation required more staff and more time, both of which are important cost drivers. Indeed, adapted sensitisation and census activities were more complicated than in settled communities as camps were not always occupied, and satellite maps needed to be reconciled with field verification visits to complete the census. Teams travelling for census or testing work had to stay several days in the field and conduct laboratory analysis at camps, increasing the number of working days and transportation costs. Finally, in addition to "settled" CDDs, nomad CDDs were employed to accompany field work, which further increased personnel costs.

Unit costs in settled communities were also relatively high compared to the economic cost of CDTi. The benchmark for Cameroon ranges from US$2.65 per person (adjusted to USD 2020 using the consumer price index) at an implementation scale of 10,000 persons, to US$0.26 for a scale of 1,000,000 persons treated [17,19]. This difference in cost is due to the TTd approach requiring more material and supplies than would otherwise be used for CDTi, and to the pilot scale and innovative nature of the implementation. In a scaled situation, we would expect that fixed costs would be reduced by the inclusion of a greater number of communities, and variable costs would be optimised through the reduction of many of the exploratory costs

and enhanced monitoring implicit in implementing innovative programming for the first time.

When comparing CDTi to TTd, it is also critical to note that TTd is curative, rather than targeting a break of transmission as CDTi does. A break of transmission requires up to 15 years of repeated programming and, in many cases where TTd is being considered, it is because long-term CDTi has not had the required impact. Initial results of an impact assessment for this implementation showed that a package of alternative treatment strategies implemented for two years in Massangam (two years of the TTd strategy, biannual CDTi and ground larviciding) had a significantly larger impact on infection prevalence in two years than the preceding 20 years of annual CDTi [9].

Given the early stage of the impact survey and the small scale of the pilot, we are not in a position to conclude on the cost-effectiveness of this approach and there is a need for further research in this area. TTd implementation could be a cost-effective intervention in regions where prevalence of microfilaria is still high, or in hard-to-reach areas where CDTi has failed to reach a high enough coverage to stop transmission or where marginalised populations consistently miss treatment. There have been reports of transmission being interrupted in certain places, but hard-to-reach foci remain that keep driving transmission [20].

In such places, TTd will result in faster progress to elimination in the whole area, instead of endless cycles of MDA. Even if the foci have low intensity of infection, transmission will be perpetuated [21]. The addition of camp verification and PCR tests will detect individuals with low infection and thus help to drive down infection. Indeed, during the second phase of the pilot, two camps and 11 additional people were found through camp verification and 161 PCR tests were conducted, detecting 14 additional infections in semi-nomadic communities, at an incremental cost of $46 per person screened (Table 5). In addition, in *loa loa* endemic areas where LoaScope is being used to exclude people from being treated with ivermectin, TTd should be considered for those being excluded [22]. This highlights how TTd could be an effective tool to address many of the challenges to onchocerciasis elimination.

Despite the rigorous financial data collection, our costing estimates suffer from several limitations. First, there were limited details provided for some of the activities–namely planning and coordination, advocacy and overheads–which meant costs could not be fully disaggregated and required the use of a mix of costing methodologies (see S1 Table). In addition, some of the planning and coordination expenditures were shared with previous rounds of implementation of TTd. For instance, the data analyst and coordinator time could not be directly attributed to the studied implementation phase and their full cost has been included, which has inflated the presented costs. Finally, and as mentioned in the methods section, the time lapse between costing work and project implementation–especially in settled communities–made it difficult to obtain solid information on opportunity cost; we have therefore decided not to estimate them.

The last mile of disease elimination is often the most difficult and the most expensive. Elimination of onchocerciasis will only be achieved if effective interventions are delivered to all people in need, including those hardest to reach. Understanding the costs of different strategies and the costs of targeting different groups–including nomadic populations–is essential for the planning and budgeting of activities and for achieving the goal of onchocerciasis elimination in Cameroon and elsewhere.

## Supporting information

**S1 Table. Methodologies by activity and sources used.** Different measurement and cost valuation methodologies were used as the level of necessary detail was not obtained to accurately

estimate the cost of several activities. S1 Table presents the methodology used to measure the cost of each activity, as well as the source of information for its valuation.
(PDF)

**S2 Table. Table of activities included in standard cost.** Table of activities included in the costing work with detailed information on expenditures required for the completion of the related implementation activity.
(PDF)

**S1 Dataset. Standard costs.** Consolidation of activities required for the completion of the test and treat strategy among settled communities and semi-nomadic communities of Massangam Health District in Cameroon. Costs were broken down to the smallest item level along with number of units required and unit costs paid for. Costs are presented in current XAF and in USD 2020.
(XLSX)

## Acknowledgments

We would like to thank Richard Selby for his support and comments on the manuscript; Alexandre Chailloux for his geographic information system expertise; the Centre de recherche sur la filariose et autres maladies tropicales (CRFilMT); the National Programme for the Control of Lymphatic Filariasis and Onchocerciasis; and the Sightsavers Cameroon office for the coordination and implementation of activities. Finally, we particularly acknowledge the communities of Massangam for their cooperation.

## Author Contributions

**Conceptualization:** Guillaume Trotignon, Ruth Dixon, Laura Senyonjo, Rogers Nditanchou.

**Data curation:** Guillaume Trotignon, Ruth Dixon, Kareen Atekem, Rogers Nditanchou.

**Formal analysis:** Guillaume Trotignon, Ruth Dixon.

**Funding acquisition:** Ruth Dixon, Rogers Nditanchou.

**Investigation:** Guillaume Trotignon.

**Methodology:** Guillaume Trotignon, Ruth Dixon, Rogers Nditanchou.

**Project administration:** Guillaume Trotignon, Laura Senyonjo, Joseph Kamgno, Didier Biholong, Iain Jones, Rogers Nditanchou.

**Resources:** Kareen Atekem, Iain Jones, Rogers Nditanchou.

**Software:** Guillaume Trotignon.

**Supervision:** Ruth Dixon.

**Validation:** Ruth Dixon, Iain Jones, Rogers Nditanchou.

**Visualization:** Guillaume Trotignon.

**Writing – original draft:** Guillaume Trotignon.

**Writing – review & editing:** Guillaume Trotignon, Ruth Dixon, Kareen Atekem, Laura Senyonjo, Joseph Kamgno, Didier Biholong, Iain Jones, Rogers Nditanchou.

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
