## [Decision Letter · Decision Letter 0]

8 Jun 2023

Dear Trotignon,

Thank you very much for submitting your manuscript "Cost of implementing doxycycline test-and-treat strategy for onchocerciasis elimination among settled and semi-nomadic groups in Cameroon" for consideration at PLOS Neglected Tropical Diseases. As with all papers reviewed by the journal, your manuscript was reviewed by members of the editorial board and by several independent reviewers. The reviewers appreciated the attention to an important topic. Based on the reviews, we are likely to accept this manuscript for publication, providing that you modify the manuscript according to the review recommendations. 

Sincerely,

Sabine Specht

Academic Editor

Nigel Beebe

Section Editor

Reviewer's Responses to Questions

**Key Review Criteria Required for Acceptance?**

**Methods**

-Are the objectives of the study clearly articulated with a clear testable hypothesis stated?

-Is the study design appropriate to address the stated objectives?

-Is the population clearly described and appropriate for the hypothesis being tested?

-Is the sample size sufficient to ensure adequate power to address the hypothesis being tested?

-Were correct statistical analysis used to support conclusions?

-Are there concerns about ethical or regulatory requirements being met?

Reviewer #1: (No Response)

Reviewer #2: Described below

Reviewer #3: - The objectives are clearly articulated with the hypothesis

- The study design is appropriate to address the stated objectives

- The population is clearly described and appropriate for the hypothesis 

- No. The sample size of the semi-nomadic population is too small to ensure adequate power to address question.

- The study did not require elaborate statistical analysis. 

- There are no concerns about ethical or regulatory requirements

**Results**

-Does the analysis presented match the analysis plan?

-Are the results clearly and completely presented?

-Are the figures (Tables, Images) of sufficient quality for clarity?

Reviewer #1: (No Response)

Reviewer #2: Described below

Reviewer #3: - Yes. The analysis presented matched the plan

- Results are clearly and completely presented

- The figures are of sufficient quality and are clear

**Conclusions**

-Are the conclusions supported by the data presented?

-Are the limitations of analysis clearly described?

-Do the authors discuss how these data can be helpful to advance our understanding of the topic under study?

-Is public health relevance addressed?

Reviewer #1: (No Response)

Reviewer #2: (No Response)

Reviewer #3: - Yes

- Yes

- Yes, the authors discussed how the outcome will stimulate further larger study

- Public Health relevance not really addressed going by the text unless the researchers took care of it but did not spell it out in the text, for example, precautions taken during the skin snips.

**Editorial and Data Presentation Modifications?**

Reviewer #1: (No Response)

Reviewer #2: • Line 57, Abstract: For readers it might be more informative to provide additional detail in the Results section, even if that requires a more succinct Conclusions section.

• Line 62, Abstract: what is ‘mf’?

• Lines 77-78: The Author Summary mentions a twice-yearly Ivermectin strategy as being included in the evaluation, but this is not mentioned in the abstract.

• Line 100: ‘experienced’ – is this the wrong word?

• Line 132: It might be useful to define ‘ground truthing’.

• Line 245-246: Seems like there is typo here (sentence fragment)? The following sentence also seems to be off. 

• Line 258: ‘From an empirical based study, financial and output data, we…’ this text is unclear. 

• Line 276: ‘(in’ Should this text be removed?

• In general: The manuscript could use copy-editing as there were several places where there appeared to be grammar issues or missing words. I have listed some below but there may be others.

Reviewer #3: Yes. These are stated in the body of the text for correction by the authors.

**Summary and General Comments**

Reviewer #1: This paper quantifies the cost of implementing a doxycycline test and-treat strategy for onchocerciasis elimination among settled and semi-nomadic groups in Cameroon. This is a novel and well presented costing study on an important area. 

My only major comment is that the perspective of the analysis needs to be explicitly stated in the methods (i.e. the costs were estimated under the perspective of _______). On a related note, the fact that opportunity costs of building use and Ministry of Public Health staff salaries were not included needs to be clearly justified in the methods in some way and this point expanded on in the limitations. 

I also believe that it would be helpful if the difference between estimated total financial and economic costs was stated.

Please state what discount rate was used for the annualization.

Reviewer #2: This manuscript describes the costs of intensified onchocerciasis control activities in rural Cameroon, where doxycycline and more frequent ivermectin distribution was used. The costing is fairly straight-forward and in general the methods seem appropriate (though note some questions/suggestions below). The writing is generally good but there are occasionally dropped words or sections where the meaning is unclear, so a careful readd-through would be helpful to resolve these. 

MAJOR ISSUES

• Line 100: The Introduction section of the manuscript contains detailed information about the intervention that I would normally expect to see in the Methods Section (e.g. under an ‘Intervention and Setting’ subheading). The authors could consider shifting this detail to the Methods section, though this may be a matter of preference / journal style.

• Line 155 (Methods): The abstract mentions that MOH staff salaries are not included, yet I cannot find this information in the Methods section. This would be important to describe here along with the activities they perform that are excluded from the costing. 

• Line 155 (Methods): It would be useful to include in the methods a clear statement of the study time frame (i.e. the period of activities being evaluated). There are several dates given in the Introduction, then different dates in line 181, so it is unclear to me right now.

• Line 195-196: Capital equipment was annualized assuming a useful life of 5 years. Conventionally these calculations include a discount rate representing the opportunity cost of capital investment. Was this done? If so the discount rate should be included here (commonly 3% or 5%). 

• Line 203: This information on discounting does not seem quite right, or may need to be edited for clarity. Ideally the costs would first be inflated to current values in local currency units, then converted to USD at current market exchange rates.

• Other comments: It would be helpful to review the CHEERS 2022 reporting framework (https://www.equator-network.org/reporting-guidelines/cheers/) to make sure all relevant items are reported. This reporting framework was built for cost-effectiveness analyses and so there will be parts of it that are not needed for a cost analysis, but still good to use. 

• The manuscript could use copy-editing as there were several places where there appeared to be grammar issues or missing words. I have listed some below but there may be others.

Reviewer #3: (No Response)

PLOS authors have the option to publish the peer review history of their article (what does this mean?). If published, this will include your full peer review and any attached files.

Reviewer #1: No

Reviewer #2: Yes: Nicolas A Menzies

Reviewer #3: No

Figure Files:

Data Requirements:

Reproducibility:

References

---

## [Decision Letter · Decision Letter 1]

18 Sep 2023

Dear Trotignon,

We are pleased to inform you that your manuscript 'Cost of implementing a doxycycline test-and-treat strategy for onchocerciasis elimination among settled and semi-nomadic groups in Cameroon' has been provisionally accepted for publication in PLOS Neglected Tropical Diseases.

Best regards,

Sabine Specht

Academic Editor

Nigel Beebe

Section Editor

Reviewer's Responses to Questions

**Key Review Criteria Required for Acceptance?**

**Methods**

-Are the objectives of the study clearly articulated with a clear testable hypothesis stated?

-Is the study design appropriate to address the stated objectives?

-Is the population clearly described and appropriate for the hypothesis being tested?

-Is the sample size sufficient to ensure adequate power to address the hypothesis being tested?

-Were correct statistical analysis used to support conclusions?

-Are there concerns about ethical or regulatory requirements being met?

Reviewer #1: (No Response)

Reviewer #3: (No Response)

**Results**

-Does the analysis presented match the analysis plan?

-Are the results clearly and completely presented?

-Are the figures (Tables, Images) of sufficient quality for clarity?

Reviewer #1: (No Response)

Reviewer #3: (No Response)

**Conclusions**

-Are the conclusions supported by the data presented?

-Are the limitations of analysis clearly described?

-Do the authors discuss how these data can be helpful to advance our understanding of the topic under study?

-Is public health relevance addressed?

Reviewer #1: (No Response)

Reviewer #3: (No Response)

**Editorial and Data Presentation Modifications?**

Reviewer #1: (No Response)

Reviewer #3: (No Response)

**Summary and General Comments**

Reviewer #1: (No Response)

Reviewer #3: I have worked on this paper earlier and given my opinion on the manuscript. If I may reiterate again, the study is reasonable, It addressed a relevant subject. I submitted that the manuscript is well written and therefore should be ACCEPTED for publication.

PLOS authors have the option to publish the peer review history of their article (what does this mean?). If published, this will include your full peer review and any attached files.

Reviewer #1: No

Reviewer #3: No

---

## [Editor Report · Acceptance letter]

13 Oct 2023

Dear Trotignon,

We are delighted to inform you that your manuscript, "Cost of implementing a doxycycline test-and-treat strategy for onchocerciasis elimination among settled and semi-nomadic groups in Cameroon," has been formally accepted for publication in PLOS Neglected Tropical Diseases.

Best regards,

Shaden Kamhawi

co-Editor-in-Chief

Paul Brindley

co-Editor-in-Chief
